# The Role of Early Revascularization and Biomarkers in the Management of Diabetic Foot Ulcers: A Single Center Experience

**DOI:** 10.3390/diagnostics12020538

**Published:** 2022-02-19

**Authors:** Ettore Dinoto, Francesca Ferlito, Manfredi Agostino La Marca, Graziella Tortomasi, Francesca Urso, Salvatore Evola, Giovanni Guercio, Marco Marcianò, David Pakeliani, Guido Bajardi, Felice Pecoraro

**Affiliations:** 1Vascular Surgery Unit, AOUP Policlinico “P. Giaccone”, 90127 Palermo, Italy; francescaferlito.ff@gmail.com (F.F.); manfredi.a.lamarca@gmail.com (M.A.L.M.); graziella.tortomasi@gmail.com (G.T.); francescaurso1992@gmail.com (F.U.); guido.bajardi@unipa.it (G.B.); felice.pecoraro@unipa.it (F.P.); 2Unit of Cardiology, Department of Health Promotion, Mother and Child Care, Internal Medicine and Medical Specialties (ProMISE) “G. D’Alessandro”, University Hospital Paolo Giaccone, University of Palermo, 90127 Palermo, Italy; cardioevola@gmail.com; 3Department of Surgical, Oncological and Oral Sciences, University of Palermo, 90127 Palermo, Italy; giovanni.guercio@unipa.it; 4Department of General and Emergency Surgery, Paolo Giaccone University Hospital, 90127 Palermo, Italy; mm.94@hotmail.it; 5Vascular Surgery Unit, Ospedali Riuniti Villa Sofia-Cervello, 90100 Palermo, Italy; davidpakeliani@gmail.com

**Keywords:** biomarkers, vascular–endovascular surgery, wound healing, tissue oxygenation, blood flow, diabetic foot ulceration

## Abstract

Diabetic neuropathy and Peripheral Arterial Disease (PAD) are the main etiological factors in foot ulceration. Herein, we report our experience of diabetic foot ulceration (DFU) management, with an analysis of the relationship between the rate of lower extremity amputation, in persons with infected DFU, after revascularization procedures performed to prevent major amputation. This study highlights the role of different biomarkers, showing their usefulness and potentiality in diabetic foot ulcer management, especially for the early diagnosis and therapy effectiveness monitoring. A retrospective analysis, from September 2016 to January 2021, of diabetic patients presenting diabetic foot with DFU, was performed. All patients were treated with at least one vascular procedure (endovascular, open, hybrid procedures) targeting PAD lesions. Outcomes measured were perioperative mortality and morbidity. Freedom from occlusion, primary and secondary patency, and amputation rate were registered. A total of 267 patients, with a mean age of 72.5 years, were included in the study. The major amputation rate was 6.2%, minor amputation rate was 17%. In our experience, extreme revascularization to obtain direct flow reduced the rate of amputations, with an increase in ulcer healing.

## 1. Introduction

Diabetic foot ulceration (DFU) is the most frequent endpoint of diabetic complications. As such, DFU represents a significant global medical, social and economic problem. It is defined as a structural or functional alteration of the foot that may manifest as ulcers, osteomyelitis, or gangrene, as a result of the interaction of different factors, induced by sustained hyperglycemia and previous traumatic causes [1,2].

This disease is expected to further increase due to the higher incidence and prevalence of type 2 diabetes registered in the last decades [3]. Diabetic peripheral neuropathy (DPN) and peripheral arterial disease (PAD) are the main etiological factors in foot ulceration. They can act alone or in combination with other factors, such as microvascular disease, biomechanical abnormalities, limited joint mobility and increased susceptibility to infection [4]. In these patients, a decrease in blood flow into the extremities, associated with damage of the microcirculation, can lead to major amputation, with an estimated incidence rate of 30.7% at 12 months [5]. Studies reporting long-term outcomes in patients presenting with DFU, estimated that the incidences of contralateral amputation and re-amputation, per 100 amputee-years in diabetic patients, are 18 and 21, respectively [6,7]. The role of biomarkers in diabetic patients with PAD has been highlighted, with particular interest on the associations of lipoprotein, inflammatory and hemostatic factors [8,9]. The position of these biomarkers in DPN and PAD has to be clarified, but the early diagnosis, with the consequent treatment and risk factors reduction, is intended to play a significant role in this important disease. The early identification of DPN and PAD is often challenging and these patients get the correct diagnosis and management when lesions are already irreversible. Thus, screening for early symptoms and signs of DPN and PAD is critical in the clinical practice, to improve patient prognosis and quality of life and reduce healthcare costs [10]. Despite the possibility to assess the main direct causes of aggressive DFU through different instrumental diagnostic methods, the difficulties related to their execution usually find a late application, when irreversible lesions are already present. In this scenario, the availability of sensitive biomarkers for the early diagnosis of DPN and PAD would be of enormous interest. The current study aims to determine the risk factors associated with lower extremity amputation in patients with infected DFU, examining the reliability of different biomarkers in different stages of the disease. The objective was to assess and test biomarkers, with the eventual predictive factor of aggressive diabetic disease in patients presenting DPN and PAD. These biomarkers have also been tested and assessed with regards to the relationship between the rate of lower extremity amputation in persons with infected DFU, after revascularization procedures performed for the prevention of major amputation with limb salvage.

## 2. Materials and Methods

The study was retrospective and included 267 diabetic patients presenting peripheral artery disease (PAD) and treated for a diabetic foot with DFU from September 2016 to January 2021. The study included patients treated with endovascular and surgical techniques during the observation period. All patients were collected and inserted into standardized piloted forms. Indications of limb salvage treatments with vascular surgery procedures were the presence of any significative vascular lesion in diabetic patients presenting DFU (Figure 1).

All the included patients gave informed consent for the procedure itself, anonymous data collection and analysis. The study was performed in agreement with the Declaration of Helsinki and the STROBE guidelines for reporting observational studies were followed [11].

The measured outcomes were perioperative mortality and morbidity. Freedom-from-occlusion, secondary patency and amputation rate were all registered. Additional maneuvers, such as surgical procedures, stenting or angioplasty with a drug eluting balloon (DEB) were reported.

The exclusion criteria were acute limb ischemia and non-atherosclerotic chronic vascular conditions of the lower extremity.

The preoperative diagnosis assessment consisted of duplex ultrasound (DUS) in all cases. Native vessel assessment parameters for DUS included peak systolic velocity (PSV), the ratio of adjacent PSVs, and phasic flow character. Computed tomography angiography (CTA) or magnetic resonance angiography (MRA) were not included in the preoperative diagnostic protocol. These radiological tests were reserved for patients presenting aorto-iliac disease.

The biomarkers analysis was performed on all patients at baseline before the first vascular procedure and after 12 months. The biomarkers analyzed were homocysteine concentration, folate, triglyceride, cholesterol (total, high-end low-density lipoprotein cholesterol and triglycerides), fasting plasma glucose, HbA1c, C-peptide, Neuropeptide Y and Elabela peptide.

The collected variables were demographics, comorbidities, clinical data, preoperative imaging studies, procedure details, type of intervention, type of anesthesia, blood transfusions, medical therapy, and length of stay. Renal function was estimated with the Chronic Kidney Disease Epidemiology Collaboration (CKD-EPI) [12].

Digital Subtraction Angiography (DSA) was used as an intraoperative diagnostic method to confirm DUS findings such as the degree of stenosis or occlusion, length of lesion, inflow and outflow assessment. The extent of arterial disease was classified according to the femoropopliteal TransAtlantic Intersociety Consensus II (TASC-II) [13,14].

The collected data were retrospectively analyzed in September 2021. The measured metrics included early technical successes (within 30 days following treatment) and late technical successes (30 days or more following treatment). Early outcomes measured included in-hospital mortality, morbidity, symptom recurrence, and amputation (major and minor). Major amputation was defined as any amputation performed above the level of the ankle, and minor amputation was defined as any amputation at the level of or below the ankle. Late outcomes included mortality, symptoms recurrence, amputation (major and minor), survival, primary patency and secondary patency. Loss of patency was calculated on a patient basis and was defined as thrombosis and/or occlusion of any treated vessel. Correlation analysis of age, comorbidities, type of treatment, blood transfusion, reinterventions, a hospital stay with complications, amputation rate, and death was performed.

Clinical follow-up consisted of a clinical examination and DUS at 1 month; after 3, 6, and 12 months; and every 6 months thereafter. The reported biomarkers were analyzed at baseline before intervention and at 12 months after the index vascular procedure. The median follow-up was 28.77 (mean: 24; r: 12–52; standard deviation [SD]: 16.43) months. For statistical analysis, means and SD or median and range were reported for parametric data; absolute values and percentages were reported for non-parametric data. Differences in preoperative and postoperative outcomes were assessed using the Student *t*-test. Kaplan–Meier curves were used to estimate survival, primary patency, and secondary patency. A bivariate test was used to assess relationship significance for correlation analysis. Statistical significance was considered at *p* < 0.05. For Kaplan–Meier curves, standard error exceeding 10% was reported. These values were log-transformed for discrete skewness. Cox regression analyses were used to investigate the association between baseline biomarker concentrations at admission and after 12 months from the revascularization procedure. We tested for linearity using a test for linear trends across the quartiles. Statistical analysis was performed using SPSS 16.0 (SPSS Inc., Chicago, IL, USA).

### 2.1. Laboratory Parameters and Biomarkers

Laboratory analyses included homocysteine, folate, triglyceride, cholesterol (total, high-end low-density lipoprotein cholesterol and triglycerides), C-peptide, fasting plasma glucose, HbA1c. These findings were analyzed and recorded from routine blood tests. Serum NPY was measured by enzyme-linked immunosorbent assay (ELISA). Serum Elabela levels were determined using commercial kits (Sunred Biological Technology, Shanghai, China).

### 2.2. Technique

Different types of procedures were performed to obtain an improvement of distal flow. The main surgical approach consisted of bypass above or below the knee (including ultra-distal bypasses) (Figure 2); the endovascular approach was used in cases of short femoro-popliteal lesion or involvement of below the knee arteries (Figure 3 and Figure 4); when a significative impairment of common femoral artery (CFA) or a multilevel vascular disease was present, and a direct surgical or endovascular procedure would never have been enough, a Hybrid Therapy solution (HT) was undertaken (Figure 5) [13].

Independently from the chosen approach, the intention to treat basis was the improvement of distal flow addressing more lesions as possible during the index treatment [15,16,17,18]. This intention to treat basis had the aim to reduce the levels of amputation turning the initial indications for major amputation into minor amputation (Figure 6).

## 3. Results

The study included 267 patients. The mean age was 72.51 (IQR: 45–80) years and 157 (58.8%) were male. Nonanatomic patient variables and medical therapy with the related grading system are reported in Table 1 and Table 2, respectively. Of the 267 patients included, 218 (82%) were diagnosed with critical limb ischemia (CLI) (Table 3). In addition to DUS, a CTA or MRA was employed as a diagnostic tool in 80 (30%) patients; the mean preoperative run-off score was 5.39 (r: 0–10; SD: 3). The values of homocysteine, triglyceride, cholesterol, blood pressure, blood glucose monitoring, Peptide C, Neuropeptide Y (NPY) and Elabela peptide were recorded (Table 4 and Table 5).

The operative management consisted of angioplasty using DEB in 154 (58%) patients, with placement of a stent in 51 (19%) cases (70–26% femoro-popliteal PTA, 84–31% BTK PTA); in 54 (20%), a bypass was employed to address PAD lesions (16–6% Iliac-femoral, 23–9% Femoro-popliteal AK, 12–5% Femoro-popliteal BK, 3–1% Femoro-tibial); in the remaining 59 (22%) patients, a hybrid procedure with endarterectomy of the common femoral artery and subsequent angioplasty in iliac and/or femora-popliteal axis (17–6% iliac axis PTA-stenting, 42–16% femoro-popliteal axis PTA) was performed.

No perioperative mortality was reported. During the follow-up, there were 11 (4%) ATK major amputations, 5 (2%) BTK major amputations, and 45 (17%) minor amputations registered. In all patients, the indication to amputation was uncontrolled foot infection. At mean follow-up, an improvement in the Rutherford clinical stage was observed, in all patients, except for the above-reported case requiring ATK major amputation.

A history of previous cardiac interventions and a larger amount of arterial segment involvement, in addition to high values of homocysteine and Elabela peptide (Table 6 and Table 7), were identified as significant risk factors for major amputation risk.

A longer hospital stay was associated with higher long-term mortality rates. Estimated 24 months survival, primary patency, secondary patency, and freedom from restenosis were 86.5%, 86.1%, 86.5% and 83.9%, respectively (Figure 7).

## 4. Discussion

Diabetes mellitus incidence and prevalence show an insidious, steady increase over time. The 2017 data released by the International Diabetes Federation Agenda, reported an estimated prevalence of 425 million people worldwide in the population from 18- to 99-year. The same Federation estimates an increase in prevalence up to almost 693 million people worldwide by 2045. The significant investments in clinical care, research, and public health interventions, showed no signs of reduction in the rate of diabetes increase [19]. Diabetic foot (DF) has a mean global prevalence of 6.4% out of the total population, and is more frequent in males than in females. Moreover, patients with DF are older, have a lower body mass index, longer diabetic duration and more hypertension, diabetic retinopathy, and smoking history than patients without DF.

The vascular disease in diabetic patients is the typical association with diabetic artery atherosclerotic multilevel diseases, involving several arterial segments and organs, with a related need to properly assess each comorbidity and complication [20,21,22]. Thus, DF disease is a significant disease, requiring multidisciplinary consensus, involving diabetologists, internists and surgeons, who should work together to coordinate revascularization procedures, aggressively, to treat infections and to manage medical comorbidities [3,23]. As reported in several reviews, for every 1% increase in hemoglobin A1c level, there is a corresponding 26% risk increase for PAD with the consequent amputation, with the risk five to ten times higher in patients with diabetes than in those without diabetes [24].

Professional workers with vascular diabetic patients are conscious that not all DFU can heal, with dramatic consequences related to amputation.

Subsequently, the need for simple diagnostic systems as biomarkers, allowing the identification of patients at higher risk of major complications is essential. Fast, non-invasive and reproducible tests, with biomarkers associated with instrumental diagnostics, may represent valid help. In the literature, several studies provide this type of correlation. Ye et al. showed a multimarker approach, based on the association between markers of hemodynamic stress (lipoprotein, inflammatory, hemostatic pathways) and the Ankle Brachial Index (ABI) [25]. In this study, higher levels of biomarkers were observed in patients presenting low ABI. In addition, biomarker monitoring can be used during the follow-up to evaluate the adequacy of therapy and the evolution of the disease. Jakubiak GK et al., in a review of the literature, about the current state of knowledge of mechanisms and the clinical significance of restenosis and in-stent restenosis (ISR), in patients with diabetes and PAD, reported an association between an elevated postoperative high-sensitivity C-Reactive Protein (hs-CRP) level, associated with an increased risk of ISR in one-year follow-up in patients who have undergone angioplasty [26]. In another study, the average hs-CRP level was shown to be significantly higher in patients in whom ISR had occurred [27]. Baktashian et al. calculated that 2.64 mg/dL was the cut-off value, below which, diabetes was the only significant factor found to predispose a patient to ISR, while when it was at least at that level, diabetes, triglyceride blood concentration, and type of revascularization treatment were factors associated with the development of ISR [28]. In a meta-analysis of six prospective observational trials, it was confirmed that the higher level of hs-CRP is associated with a significantly increased risk of ISR (OR 1.16, 95% CI 1.01–1.30; *p* < 0.05) [29]. Olinic DM et al. reported elevated CRP levels as a predictor for symptomatic PAD development over the next five years, in former asymptomatic subjects [2,24,30].

The latest literature has revealed the role of interesting biomarkers, such as the Neuropeptide Y (NPY) and the Elabela Peptide. Cho et al. showed that high levels of this marker are visible from the earlier times of diabetic disease, before early symptoms [10]. Elabela peptide is a product of the renin-angiotensin-aldosterone system; Kaplan et al. showed significant growth of this marker in patients with high grades of Rutherford and WIfi scale, with an increase in cardiovascular disease risk [31]. Passaro et al. reported a study on 95 patients, where the homocysteine concentration, a biomarker known in the literature for its close link with vascular disease, is reduced by an improved metabolic control [32].

In our patients, the same biomarkers have confirmed the trend characterized by high values in the categories with major vascular impairment. In addition, the analysis of our results suggests that C-peptide, the product of insulin metabolism, is a possible valid biomarker that shows the proper functioning of blood glucose levels overall in patients with renal failure [33].

As described in Table 3, the improvement of peripheral flow, after the surgical or endovascular treatment was associated with clinical recovery, with a reduction in the inflammatory state and a more effective impact of antibiotic therapy, with a decrease in the values of the biomarkers examined. This trend could be related to a minor inflammatory state, a condition at the basis of both vascular disease and lesions of the diabetic foot.

Herein, it is evident that diabetic arterial disease requires a different approach when compared to peripheral arterial disease. Such interventions aim to increase the blood flow to the foot, which in turn, enhances cutaneous oxygen pressure, promoting infection clearance and ulcer granulation [34,35,36]. Patients affected by diabetic artery disease often evolve toward critical limb-threatening ischemia (CLTI), with a multilevel arterial involvement (multilevel peripheral artery disease—MPAD) of up to 90% [37,38,39]. To achieve consistent clinical improvements, these patients require extensive multilevel reconstruction [40,41,42]. Aggressive revascularization with a surgical or endovascular approach is accepted in order to preserve functional limbs, permit pain relief, wound healing and improve quality of life.

However, there is still a lack of evidence, especially from randomized clinical trials, as to whether an endovascular-first approach provides a benefit over conservative treatment and/or surgery of DF ulcers in the above the knee district, while an endovascular-first strategy is confirmed to have a prominent role in cases of the below the knee disease. An extensive and complex surgical approach is still indicated in MPAD, but endovascular solutions have been reported to reduce the invasiveness of conventional surgery [43,44,45]. However, the only endovascular approach, or a direct surgical solution, is often technically inadequate to address simultaneous MPAD. In these cases, hybrid treatments can represent a valuable option in patients considered at high risk for conventional surgery.

The most common complication of diabetic arterial disease is undoubtedly amputation. Incidence of lower extremity amputation ranges from 5.8–31 per 105 in the general population to 46.1–9600 per 105 in the diabetic population, of which 85% of amputations follow an ulcer complicated by gangrene and infection. Timing in revascularization (‘‘time is tissue’’) is a crucial aspect for the patient outcome; an early restoration of blood flow, coupled with an extensive surgical debridement, lowers mortality, major amputation, and enhances foot healing [46,47,48,49]. In this study, the analysis of metabolic characteristics showed the importance of appropriate therapy, understood as a union of drugs and revascularization treatments. In fact, in all patients, an increase in the measured biomarkers, before the revascularization procedure, was observed. At the follow-up control, at 12 months from the index procedure, improvement of the measured biomarkers was observed in patients presenting an efficient revascularization, in association with consistent metabolic control. By contrast, patients undergoing major amputation, after failed revascularization procedures and/or inefficient metabolic control, did not present an improvement in the measured biomarkers.

In our experience, an aggressive revascularization behavior showed that obtaining an improvement of distal flow allowed us to reduce the levels of amputation, turning major amputations to minor amputations. The choice of the approach is dependent on the anatomical characteristics of arterial lesions and is in accordance with the clinical status and based on ulcer degree. A higher incidence of the endovascular approach has guaranteed the use of less invasive procedures with better outcomes.

The current study presents several limitations, including the lack of a comparison control group and a relatively small patient sample size. Although this is a prospective study, the lack of randomization and the use of different revascularization techniques represent further limitations. It is not to be neglected that this monocentric study has been conducted without any industry sponsorship.

## 5. Conclusions

Extreme revascularization to achieve distal direct flow reduces the rate of amputation and increases ulcer healing. The reported study shows better outcomes in terms of limb salvage and amputation rate reduction, when a vascular treatment is performed at an early stage of diabetic foot ulceration and can improve distal trophism. To assess early diabetic peripheral arterial disease, a multi-biomarkers approach can be employed to identify, early, patients at higher risk of disease progression or inefficient therapy. The frequent multilevel localization of peripheral disease should be aggressively managed, despite the higher risk of comorbidities and complications. A standardized and multidisciplinary approach to the DFU is necessary to improve outcomes, such as amputation-free survival and mortality.

## Figures and Tables

**Figure 1 diagnostics-12-00538-f001:**
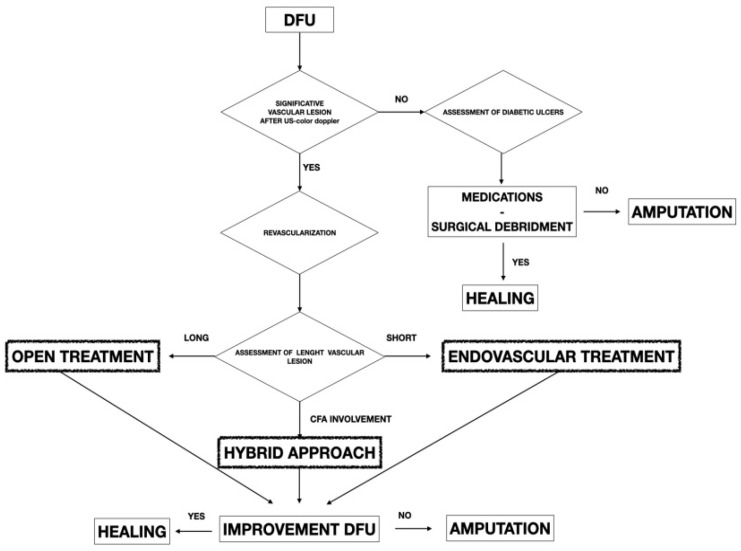
Flowchart showing the decisional algorithm employed in patients presenting PAD with DFU.

**Figure 2 diagnostics-12-00538-f002:**
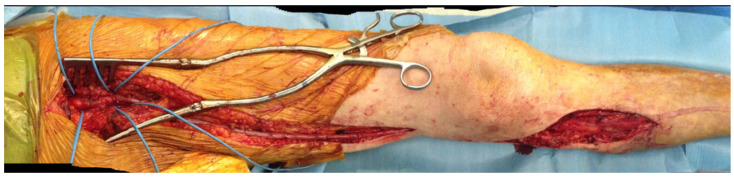
Final outcome of autologous inverted saphenous femoro-popliteal BTK bypass.

**Figure 3 diagnostics-12-00538-f003:**
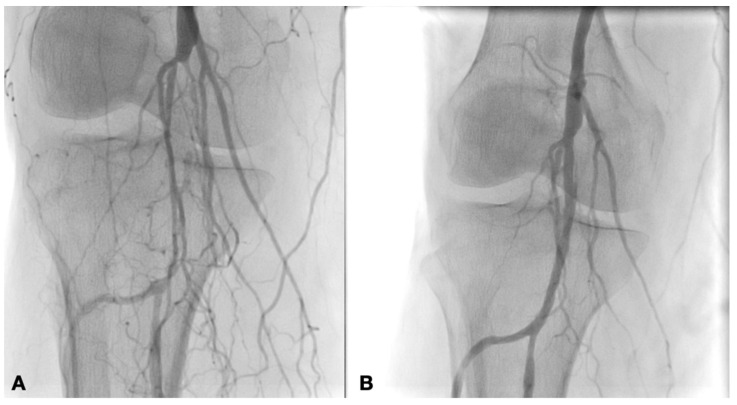
Intraoperative angiography showing popliteal artery before (**A**) and after PTA with Drug Eluting Balloon (**B**).

**Figure 4 diagnostics-12-00538-f004:**
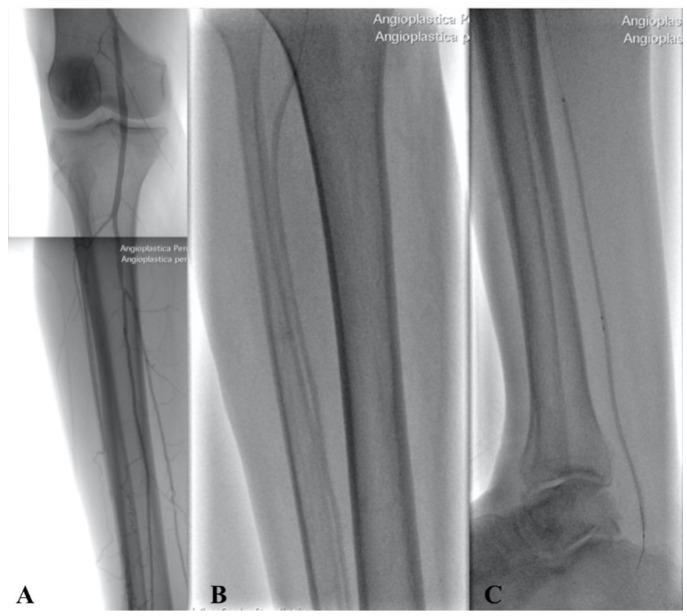
Intraoperative angiography showing popliteal and BTK arteries after PTA (**A**) with Balloon in Anterior Tibial Artery (**B**) and Posterior Tibial Artery (**C**).

**Figure 5 diagnostics-12-00538-f005:**
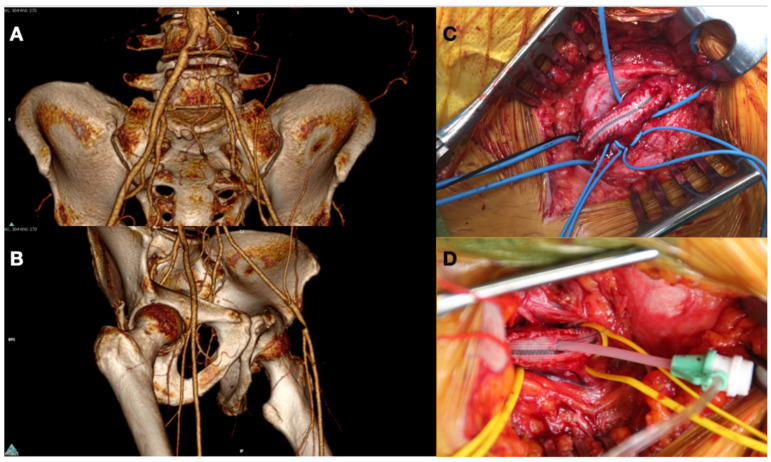
Preoperative CT Angiography showing occlusion in left Common Iliac Artery (**A**) and critical stenosis in Common Femoral Artery (**B**). Puncture of Common Femoral Artery after endarterectomy and Dacron patch angioplasty (**C**) with placement of sheath to perform a treatment of iliac lesion (**D**).

**Figure 6 diagnostics-12-00538-f006:**
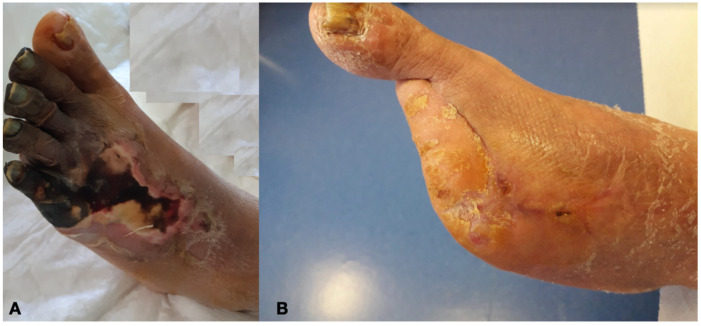
Diabetic Foot before (**A**) and after (**B**) revascularization with transmetatarsal amputation.

**Figure 7 diagnostics-12-00538-f007:**
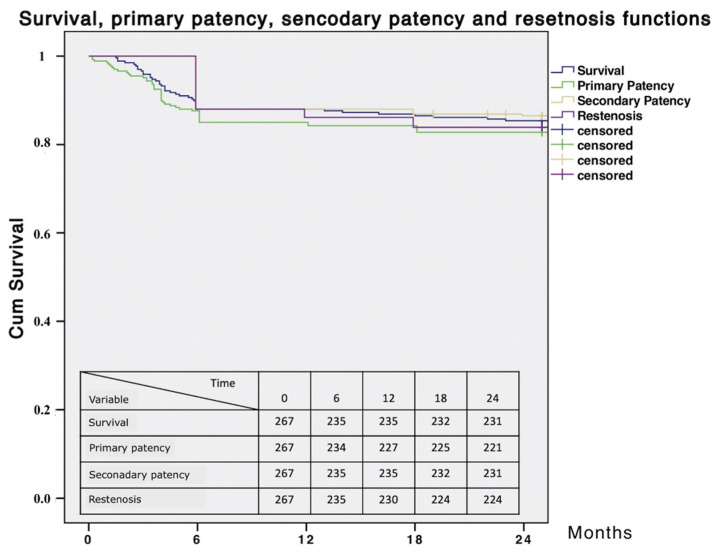
Survival, primary patency, secondary patency and restenosis curves.

**Table 1 diagnostics-12-00538-t001:** Nonanatomic patient variables.

Categories	Grade	N	%
Diabetes	None	0	0
	Not rerquiring insulin	40	15
	Controlled by insulin	199	75
	Type 1 or uncontrolled	28	10
Tobacco Use	None (>10 years ago)	70	26
	Quit 1–10 bears ago	45	17
	Current within last year, <1 package per day	113	42
	Current within last year, >1 package per day	39	15
Hypertension	None	15	6
	Controlled with 1 drug	66	25
	Controlled with 2 drugs	126	47
	Requiring > 2 drugs or uncontrolled	60	22
Renal Status	Normal	86	32
	Evidence of renal disease, GFR > 90 mL/min/1.73 m^2^	77	29
	GFR 60–89 mL/min/1.73 m^2^	41	15
	GFR 30–59 mL/min/1.73 m^2^	43	16
	GFR 15–29 mL/min/1.73 m^2^	5	2
	GFR < 15 mL/min/1.73 m^2^	15	6
Cardiac status	Asymptomatic	87	32
	Asymptomatic, but with remote myocardial infarction by history (6 months)	72	27
	Stable angina, ejection fraction 25% to 45%, controlled ectopy or asymptomatic arrhythmia, or history of congestive heart failure that is now well compensated	76	28
	Unstable angina, ejection fraction <25%, myocardial infarction ≤6 months	32	12
Functional status	No impairment	15	6
	Impaired, but able to carry out ADL without assistance	49	18
	Needs some assistance to carry out ADL or ambulatory assistance	116	44
	Requiring total assistance for ADL or nonambulatory	87	32
ADL, Activities of daily living; FEV1, forced expiratory volume in 1 s; GFR, glomerular filtration rate			

**Table 2 diagnostics-12-00538-t002:** Medical Therapy before intervention.

Categories	Grade	N	%
Antiplatelet Therpay	None	64	24
	Single Agent	155	58
	Dual Therapy	48	18
Lipid-lowering therapy	On statin and lipid testing within therapeutic range	187	70
	On Statin but lipid levels not optimal	48	18
	On high-dose statin or requiring supplemental lipid medications	15	6
	Statin intolerant or patient with familial hypercholesterolemia	17	6
Anticoagulation	No Hypercoagulable State	242	91
	Unknow	2	1
	Low risk Hypercoagulable state	0	0
	High risk Hypercoagulable state	23	8

**Table 3 diagnostics-12-00538-t003:** Clinical presentation and anatomic findings.

Categories	Grade	N	%
Rutherford Classification	3	50	19
	4	78	29
	5	110	41
	6	29	11
WIfi Classification	Clinical Stage 1	34	13
	Clinical Stage 2	29	11
	Clinical Stage 3	50	22
	Clinical Stage 4	145	54
TASC Classification	TASC B	72	27
	TASC C	131	49
	TASC D	64	24

**Table 4 diagnostics-12-00538-t004:** Metabolic Characteristics.

Variable	Time 0	12 Months
Serum Homocisteine (µmol/L)	11.7 ± 2.1	11.5 ± 1.8
Serum Folate (nmol/L)	22 ± 6	28 ± 5
Plasma Triglycerides (mg/dL)	183 ± 21	151 ± 18
Plasma Cholesterol (mg/dL)	221 ± 6	190 ± 11
LDL-Cholesterol (mg/dL)	185 ± 6	90 ± 6
HDL-Cholesterol (mg/dL)	10 ± 7	23 ± 6
Fasting plasma glucose (mg/dL)	221 ± 53	153 ± 12
HbA1c (%)	8.4 ± 1.2	7.5 ± 1.8
C-peptide (ng/mL)	1.9 ± 1.2	0.9 ± 1.5
Neuropeptide Y (pg/mL)	2983 ± 88	2970 ± 95
Elabela peptide(ng/mL)	2.9 ± 2.1	1.8 ± 1.9

**Table 5 diagnostics-12-00538-t005:** Clinical presentation and Biomarkers.

	Variable	Time 0	12 Months
Rutherford Classification			
Category 3	Serum Homocisteine (µmol/L)	10.7 ± 2.1	9.5 ± 0.4
	Fasting plasma glucose (mg/dL)	150 ± 21	145 ± 1
	C-peptide (ng/mL)	1.9 ± 0.5	0.9 ± 0.2
	Neuropeptide Y (pg/mL)	2933 ± 38	2930 ± 20
	Elabela peptide(ng/mL)	2.8 ± 1	1.5 ± 1.4
Category 4	Serum Homocisteine (µmol/L)	11.8 ± 2.0	11.0 ± 1.2
	Fasting plasma glucose (mg/dL)	185 ± 12	154 ± 7
	C-peptide (ng/mL)	11.8 ± 2.0	11.0 ± 1.2
	Neuropeptide Y (pg/mL)	2.0 ± 0.2	0.9 ± 0.5
	Elabela peptide(ng/mL)	2968 ± 23	2945 ± 22
Category 5	Serum Homocisteine (µmol/L)	11.8 ± 2.2	11.4 ± 1.7
	Fasting plasma glucose (mg/dL)	191 ± 10	161 ± 3
	C-peptide (ng/mL)	2.3 ± 0.5	1.8 ± 0.5
	Neuropeptide Y (pg/mL)	3001 ± 12	2940 ± 44
	Elabela peptide(ng/mL)	3.3 ± 1.3	2.7 ± 0.5
Category 6	Serum Homocisteine (µmol/L)	12.3 ± 2.1	12.5 ± 1.7
	Fasting plasma glucose (mg/dL)	205 ± 18	210 ± 3
	C-peptide (ng/mL)	2.4 ± 0.6	2.4 ± 0.7
	Neuropeptide Y (pg/mL)	3003 ± 15	3002 ± 41
	Elabela peptide(ng/mL)	3.5 ± 1.2	3.5 ± 0.2
WIfi Classification			
Clinical Stage 1	Serum Homocisteine (µmol/L)	11.6 ± 2.2	11.4 ± 1.3
	Fasting plasma glucose (mg/dL)	151 ± 23	147 ± 3
	C-peptide (ng/mL)	1.8 ± 0.5	0.8 ± 0.3
	Neuropeptide Y (pg/mL)	2943 ± 28	2935 ± 19
	Elabela peptide (ng/mL)	2.7 ± 0-9	1.6 ± 1.3
Clinical Stage 2	Serum Homocisteine (µmol/L)	11.9 ± 1.1	11.2 ± 1.1
	Fasting plasma glucose (mg/dL)	187 ± 9	164 ± 3
	C-peptide (ng/mL)	1.9 ± 0.2	1.1 ± 0.4
	Neuropeptide Y (pg/mL)	2953 ± 32	2951 ± 12
	Elabela peptide(ng/mL)	2.8 ± 1.7	2.2 ± 0.2
Clinical Stage 3	Serum Homocisteine (µmol/L)	12.1 ± 1.4	11.5 ± 1.3
	Fasting plasma glucose (mg/dL)	188 ± 13	169 ± 9
	C-peptide (ng/mL)	2.2 ± 0.7	1.9 ± 0.4
	Neuropeptide Y (pg/mL)	3014 ± 21	2990 ± 24
	Elabela peptide(ng/mL)	3.1 ± 0.3	2.55 ± 0.5
Clinical Stage 4	Serum Homocisteine (µmol/L)	12.7 ± 1.1	11.6 ± 2.1
	Fasting plasma glucose (mg/dL)	210 ± 15	181 ± 30
	C-peptide (ng/mL)	2.4 ± 0.6	1.9 ± 0.7
	Neuropeptide Y (pg/mL)	3012 ± 12	2980 ± 63
	Elabela peptide(ng/mL)	3.3 ± 1.0	2.8 ± 0.2

**Table 6 diagnostics-12-00538-t006:** Variables in patient with major amputation (16 patients).

Categories	Grade	N	%
Diabetes	None	0	0
	Not 12 equiring insulin	0	0
	Controlled by insulin	16	100
	Type 1 or uncontrolled	0	0
Tobacco Use	None (>10 years ago)	0	0
	Quit 1–10 bears ago	2	12
	Current within last year, <1 package per day	8	50
	Current within last year, > 1 package per day	6	38
Hypertension	None	0	0
	Controlled with 1 drug	10	62
	Controlled with 2 drugs	5	32
	Requiring > 2 drugs or uncontrolled	1	6
Renal Status	Normal	4	26
	Evidence of renal disease, GFR > 90 mL/min/1.73 m^2^	3	19
	GFR 60–89 mL/min/1.73 m^2^	2	12
	GFR 30–59 mL/min/1.73 m^2^	3	19
	GFR 15–29 mL/min/1.73 m^2^	2	12
	GFR < 15 mL/min/1.73 m^2^	2	12
Cardiac status	Asymptomatic	2	12
	Asymptomatic, but with remote myocardial infarction by history (6 months)	6	38
	Stable angina, ejection fraction 25% to 45%, controlled ectopy or asymptomatic arrhythmia, or history of congestive heart failure that is now well compensated	7	44
	Unstable angina, ejection fraction <25%, myocardial infarction ≤6 months	1	6
Patency BTK vessels before procedure	3	0	0
	2	2	12.5
	1	12	75
	0	2	12.5

**Table 7 diagnostics-12-00538-t007:** Metabolic Characteristics in patient with major amputation (16 patients).

Variable	
Serum Homocisteine (µmol/L)	13.7 ± 1.1
Serum Folate (nmol/L)	17 ± 5
Plasma Triglycerides (mg/dL)	223 ± 11
Plasma Cholesterol (mg/dL)	251 ± 7
LDL-Cholesterol (mg/dL)	195 ± 5
HDL-Cholesterol (mg/dL)	9 ± 4
Fasting plasma glucose (mg/dL)	271 ± 35
HbA1c (%)	9.4 ± 2.2
C-peptide (ng/mL)	1.7 ± 0.2
Neuropeptide Y (pg/mL)	3421 ± 75
Elabela peptide(ng/mL)	3.1 ± 1.2

## Data Availability

The data presented in this study are available on request from the corresponding author. The data are not publicly available due to privacy.

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
