# Peer review of "The Role of Early Revascularization and Biomarkers in the Management of Diabetic Foot Ulcers: A Single Center Experience"

_diagnostics, 2022, doi:10.3390/diagnostics12020538_

Round 1
Reviewer 1 Report
Please revise spelling, syntax and grammar. Maybe professional editing by native speaker will clarify the paper.
Introduction misses built-up to research question, no literature support for biomarkers. What is the hypothesis?
The discussion is not a discussion but a continuation of the introduction.
Author Response
Introduction misses built-up to research question, no literature support for biomarkers. What is the Hypothesis
The study aims to determine the risk factors associated with lower extremity amputation in patients with infected DFU examining the reliability of different biomarkers in different stages of the disease. The objective was to assess and test biomarkers with the eventual predictive factor of aggressive diabetic disease in patients presenting DPN and PAD. The literature pertaining to this topics is reported in the discussion.
The discussion is not a discussion but a continuation of the introduction
I improved the discussion but I think that this is complete with references to literature, commentated and compared to our outcomes.
Reviewer 2 Report
The current manuscript is well written and focused on an interesting topic. I would suggest to the authors to report the accuracy of choosen biomarkers in predicting the outcomes of interest. They could rely on C-index for survival outcomes and on AUC by ROC curves for binomial outcomes.
Author Response
The current manuscript is well written and focused on an interesting topic. I would suggest to the authors to report the accuracy of choose biomarkers in predicting the outcome of interest. They could rely on C-index for survival outcomes and on AUC by ROC curves for binomial outcomes.
I thank you for this positive comment. I improved the statistical analysis but I have not entered a ROC curve because I consider this not suitable
Reviewer 3 Report
The topic of this manuscript falls within the scope of Diagnostics
The study was planned to be a retrospective and analysis including 267 diabetic patients presenting peripheral artery disease (PAD) and treated for diabetic foot with DFU. The aims of the study were to determine the risk factors associated with lower extremity amputation in patients with infected DFU examining the reliability of different biomarkers in different stages of disease. The Authors showed that extreme revascularization in search of direct flow reduced the rate of amputations with an increase in ulcer healing. The study presented better outcomes in term of limb salvage and amputation rate reduction when a vascular treatment is performed at early stage of diabetic foot ulceration and it is able to improve the distal trophism.
The strength of this paper are: very interesting and important topic, which gives us new information about diabetic foot with ulcer. This knowledge is necessary during the treatment of diabetic foot; material and methods-the right choice of methodology methods, which was presented in comprehensible way; the obtained results are presented in the form of figures, which are clear and easy to understand; the conclusions- based on the obtained results;
There are some comments in the reviewer opinion which should be taken under consideration by the Authors:
- In the introduction or discussion please cite the newest papers in this field:
- doi: 10.3390/ijerph182211970. PMID: 34831726
- doi: 10.2991/jegh.k.191028.001. PMID: 32175717;
- doi: 10.1155/2019/6036359. PMID: 31049356;
- doi: 10.23736/S0392-9590.18.03996-2
- doi: 10.1186/s12933-019-0955-5. PMID: 31730004
- doi: 10.5312/wjo.v12.i2.61. PMID: 33614425
- Please add limitations of your study
- What about the patients with diabetic foot and ulcer, who had infections. It was presented in the table 3 WIfi Classification. Could you discuss your obtained results depending on the inflammation? Whether the levels of biomarkers correlated with WIfI classification? [doi: 10.1016/j.jvs.2018.01.060; PMID: 29803684.]
Author Response
Thanks for the advice. I followed your indications

Round 2
Reviewer 1 Report
The paper hasn't changed. There are spelling errors in the changes.
Author Response
I tried to improve my paper

This manuscript is a resubmission of an earlier submission. The following is a list of the peer review reports and author responses from that submission.